

# Spatially explicit models of density improve estimates of Eastern Bering Sea beluga (*Delphinapterus leucas*) abundance and distribution from line-transect surveys

Megan C. Ferguson[1,2], Paul B. Conn[1] and James T. Thorson[3]

[1] Marine Mammal Laboratory, Alaska Fisheries Science Center, National Marine Fisheries Service, National Oceanic and Atmospheric Administration, Seattle, WA, United States of America

[2] Biodiversity Research Institute, Portland, ME, United States of America

[3] Resource Ecology and Fisheries Management, Alaska Fisheries Science Center, National Marine Fisheries Service, National Oceanic and Atmospheric Administration, Seattle, WA, United States of America

Corresponding author
Megan C. Ferguson,
megan.ferguson@briwildlife.org

## ABSTRACT

We investigate spatially explicit models and ensemble modeling techniques for estimating animal abundance from line-transect survey data. Spatially explicit models are expected to be statistically more efficient, resulting in more precise abundance estimates, than design-based abundance estimators that rely heavily on assumptions about survey design and realization. Ensemble modeling reduces error by averaging among models, and allows for model selection uncertainty to propagate to the abundance estimator. We develop density surface models using Matérn covariance functions and spline-based smooths for a case study, belugas (*Delphinapterus leucas*) from the Eastern Bering Sea (EBS) stock. EBS belugas are upper trophic level predators in a rapidly changing ecosystem and are a vital nutritional and cultural resource for Alaska Natives. Effective management of this stock requires regular monitoring to derive accurate and unbiased estimates of abundance. Since 1992, aerial line-transect surveys have been the primary means of surveying and estimating abundance of EBS belugas in the region. We compare EBS beluga abundance estimates for 2017 and 2022 that were derived using post-stratified, design-based abundance estimators with analogous estimates the we derive using spatially explicit and ensemble modeling methods. The estimated precision in the abundance estimates from the individual density surface models (DSMs) and the ensemble average of DSMs is higher than for the design-based estimator in both survey years. The design-based models estimated that there were 12,269 belugas in 2017 (coefficient of variation (CV) = 0.118) and 19,811 belugas within a larger study area in 2022 (CV = 0.343). The ensemble spatial models estimate that there were 11,654 belugas in 2017 (CV = 0.118) and 13,313 belugas in 2022 (CV = 0.216). Among the individual spatially explicit models, abundance estimates range from 11,242 to 11,963 (CV = 0.111 to 0.114) in 2017 and 12,023 to 15,593 (CV = 0.172 to 0.198) in 2022. Because spatial models identify spatial patterns in beluga density at finer resolutions than design-based models, we argue that ensembles of spatially explicit density models provide a reasonable path forward for estimating EBS beluga abundance and distribution in a way that is useful to management and conservation efforts.

**Subjects** Conservation Biology, Natural Resource Management, Spatial and Geographic
Information Science
**Keywords** Abundance estimation, Spline-based smoother, SPDE, Density surface model,
Ensemble model, Uncertainty estimation

# INTRODUCTION

Natural resource management inevitably involves choosing among alternative actions
that may have different effects on a population in the future. Therefore, effective wildlife
management is often based on estimates of abundance, together with a characterization
of uncertainty. Our ability to predict the future depends on how well we know the
ecosystem today and the magnitude and direction of cascading effects that may result from
a particular management action. Transparent communication about scientific uncertainty
is particularly important when managing populations that are hunted for Native subsistence
due to the animals' nutritional, cultural and spiritual value to Indigenous peoples.

For decades, line-transect data were analyzed using a two-stage process invoking model-
based inference to estimate detection probability, followed by design-based inference to
extrapolate an estimate of the number of animals on the surveyed transects to an estimate of
the number of animals throughout the study area. Design-based inference has a rich history
in sampling (*Cochran, 1977*), and is appealing in its simplicity. In particular, random or
systematic placement of transects ensures that simple extrapolations of densities from
sampled to unsampled areas (*e.g.*, using simple random sampling or stratified random
sampling estimators) will be unbiased, assuming that the specified design was correctly
followed during field sampling.

In the context of line-transect sampling, spatially explicit, model-based estimators
are often referred to as "density surface models" (DSMs; *Miller et al., 2013*). Since the
early 2000s, model-based approaches to inference from line-transect survey data (*Hedley
& Buckland, 2004*; *Johnson, Laake & Ver Hoef, 2010*; *Miller et al., 2013*; *Yuan et al., 2017*)
have seen increased use relative to design-based inference. Modeling animal density as a
function of spatial or environmental covariates may increase precision and reduce bias
in the overall abundance estimate for the survey area. This applies particularly to cases
in which animal density is spatially heterogeneous and achieved survey coverage is non-
uniform, for example, due to incomplete survey effort or spatially heterogeneous detection
probability (*Hedley & Bravington, 2014*). Additionally, DSMs can be used to create high-
resolution maps of animal density, which are useful for marine spatial planning, estimating
potential impacts from anthropogenic activities, and investigating ecological relationships.
We are particularly interested in the sensitivity of abundance estimates to DSM model
structure, how this variance propagates through ensemble models, and how model-based
abundance estimators compare with conventional post-stratified design-based abundance
estimators. We identify similarities and differences among different analytical approaches
both theoretically and with a case study, the Eastern Bering Sea (EBS) beluga whale
(*Delphinapterus leucas*) stock, which is hunted for subsistence by Alaska Natives.

EBS belugas are vital to Indigenous communities near Norton Sound and the Yukon
River Delta in northwestern Alaska (Fig. 1). The northern Bering Sea ecosystem is

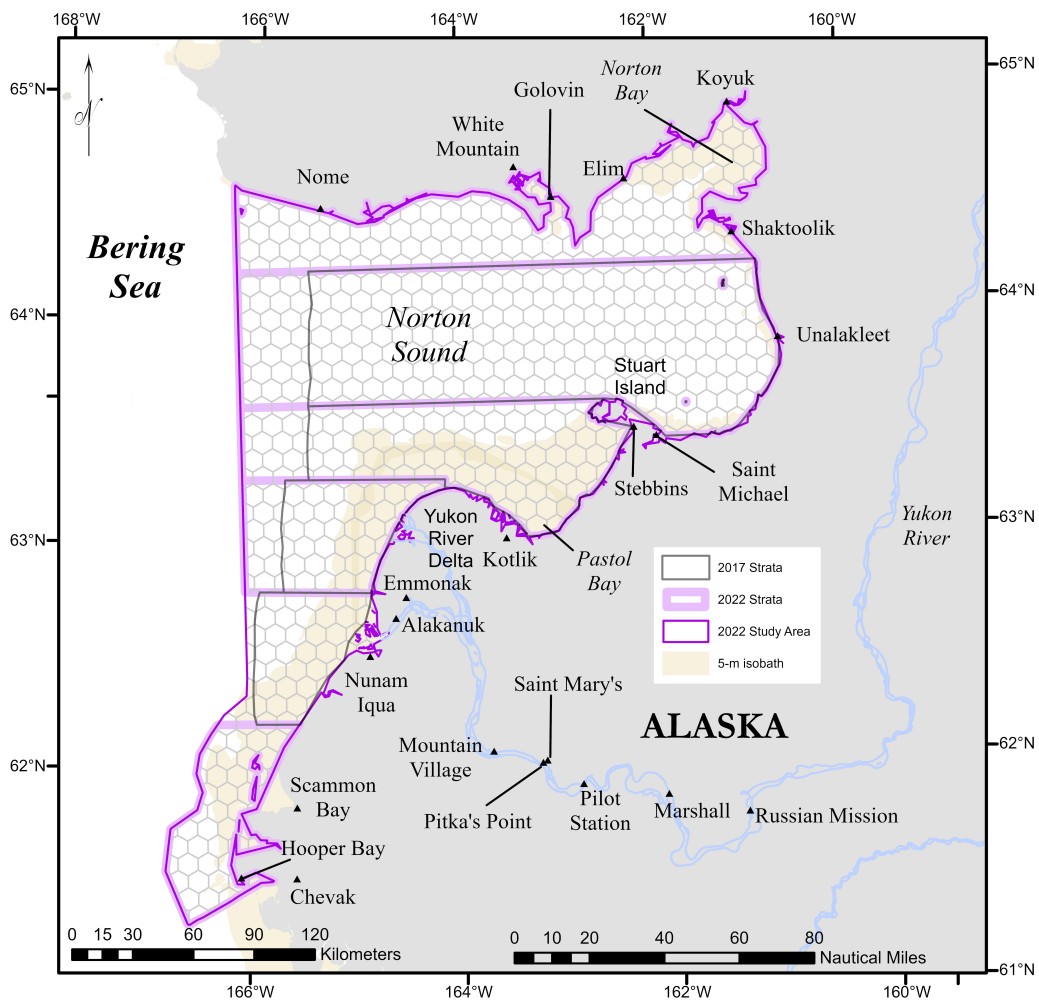

**Figure 1** **Study area for the Eastern Bering Sea beluga case study.** The geographic strata used in the design-based abundance estimates for 2017 and 2022 are outlined in dark gray and lavender, respectively. The complete 2022 study area is outlined in violet.

experiencing rapid ecological changes (*Siddon, 2023*) and increased human activities. EBS belugas are one of four beluga stocks that have been co-managed since 1988 by the Alaska Beluga Whale Committee (ABWC) and the U.S. National Oceanic and Atmospheric Administration (NOAA)/National Marine Fisheries Service (NMFS) (*Adams, Frost & Harwood, 1993*; *Frost et al., 2021*). The ABWC includes hunters, resource managers, and scientists. The goals of the ABWC are to maintain healthy beluga populations in Alaska waters, provide adequate subsistence harvest of beluga whales, and protect hunting privileges for Alaskan subsistence hunters (*Frost et al., 2021*). Since its founding, the ABWC has believed that education, maintaining accurate harvest data and conducting surveys to estimate stock abundance on a regular schedule are critical to the health of northwestern Alaska's beluga stocks and the communities that depend on them.

The distribution and movement patterns of EBS belugas are primarily known from Indigenous and other local knowledge (*Huntington & Communities of Buckland, Elim, Koyuk, Point Lay, and Shaktoolik, 1999*; *Oceana and Kawerak Inc., 2014*; *Lowry et al., 2017*), aerial surveys (*Lowry et al., 2017*; *Ferguson et al., 2023*), telemetry studies (*Citta et al., 2017*), and genetics (*e.g.*, *O'Corry-Crowe et al., 2018*; *O'Corry-Crowe et al., 2021*). EBS belugas predictably occur in the Norton Sound/Yukon Delta region during the period from shortly after sea ice breakup (usually mid-May) until freeze-up (usually November) (*Lowry et al., 2017*; *Citta et al., 2017*). Belugas from this stock are hunted by more than 20 villages during spring, summer and autumn (*Lowry et al., 2019*). EBS beluga distribution from spring through autumn reflects high densities of prey, particularly fishes (*Lowry et al., 2017*), but can also be affected by sea ice conditions and human disturbance (*Huntington & Communities of Buckland, Elim, Koyuk, Point Lay, and Shaktoolik, 1999*; *Oceana and Kawerak Inc., 2014*).

To obtain an estimate of EBS beluga abundance that could be used to evaluate the sustainability of beluga subsistence harvests, the ABWC conducted aerial surveys in Norton Sound and along the Yukon River Delta each year from 1992 to 1995 and 1999 to 2000 (*Lowry et al., 2017*). In 1992, aerial surveys were conducted in May, June and September to determine the best month for conducting future surveys. Based on those results, aerial surveys for all remaining years were conducted in June, when belugas tend to concentrate near Pastol Bay and the Yukon River Delta (Fig. 1). *Lowry et al. (2017)* estimated EBS beluga abundance to be 6,994 belugas (95% confidence interval 3,162–15,472) based on the aerial surveys conducted in June 2000. This estimate included a correction factor of 2.0 to account for availability bias (*Marsh & Sinclair, 1989*), which arises when belugas in the area searched during the surveys are underwater when the aircraft flies over them.

During June 2017 and 2022, ABWC and NMFS collaborated to conduct aerial line-transect survey in the Norton Sound/Yukon Delta region to collect data to derive updated abundance estimates for the EBS beluga stock. *Ferguson et al. (2023)* presented an estimate of EBS beluga abundance of 12,269 belugas (coefficient of variation (CV) = 0.12) based on the 2017 surveys. Their abundance estimate incorporated correction factors for availability bias and transect detection probability, and was derived using design-based methods with post-stratification (*Ferguson et al., 2023*). Compared to 2017, the 2022 surveys included less survey effort and beluga sightings were more patchily distributed. We were interested in whether DSMs could provide a reasonable alternative to design-based abundance estimators for these two most recent survey years. There are a considerable number of ways to formulate DSMs. Therefore, we also examined how different types of DSMs performed on the same dataset. We expected to see differences among density surfaces and total population abundance estimates across DSMs due to differing assumptions about spatial covariance.

This paper compares different analytical methods used to estimate population abundance from line-transect survey data for the purpose of effectively managing a population. The remainder of this paper is structured as follows: (i) introduction to basic estimators of animal density and abundance from line-transect survey data; (ii) definition of the marginal likelihood that forms the core of the DSMs; (iii) definition of the random effects

that form the basis of the different DSMs that we compared; (iv) description of methods for predicting abundance from individual DSMs; (v) explanation of model validation and evaluation methods; (vi) description of methods for calculating uncertainty in abundance estimates for each individual DSM; (vii) definition of the ensemble modelling approach that was used to account for model selection uncertainty; and, finally, (viii) application to the EBS beluga case study, focusing on how results from the individual DSMs, ensemble DSMs, and design-based estimators of abundance compare. Throughout, we assume that the reader is familiar with basic terminology and definitions associated with distance sampling (*Buckland et al., 2001*).

## MATERIALS & METHODS

Fundamentally, our density surface model uses assumptions about the spatial relationships among animals in a particular geographic area to extrapolate from what is known about the number and distribution of animals sighted on transects during a survey to an estimate of the total number of animals that were truly present in the geographic area during the survey period. We estimate abundance independently for a specified point or period in time and do not explicitly model changes in abundance over time. A density surface represents the estimated density (number of individuals per unit area) of animals in each cell of a grid. To estimate total abundance during a given survey period, we integrate across the density surface, which involves multiplying each cell's estimate of animal density by its geographic area and then summing cellwise abundances across all cells in the study area.

The analytical methods that we present below may be used for a wide range of datasets and taxa. However, to employ them one will generally need access to (1) a line-transect observation dataset to estimate a detection function and animal density; (2) auxiliary information to estimate detection probability at distance zero; and (3) auxiliary information to estimate availability probability. To understand the flexibility in the methods and critical elements that were included to accommodate the *Eastern Bering Sea beluga case study*, we note that three independent datasets were used in the case study: (1) aerial line-transect marine mammal observer (*i.e.,* "aerial observer") data from the eastern Bering Sea in 2017 and 2022 (Supplement 2, hereafter "S2") were used to estimate a multiple covariates distance-sampling (MCDS) detection function (S4) and to construct the DSMs; (2) aerial imagery collected in the eastern Chukchi and western Beaufort seas during July through October in 2018 and 2019 (S3) were used to estimate the probability of detecting a beluga group on the transect line (*Buckland et al., 2015*; *Laake & Borchers, 2004*; S4); and (3) Very High Frequency (VHF) telemetry data from Bristol Bay, Alaska, in June 1983, and Cunningham Inlet, Somerset Island, Canada, in July 1988 (*Frost, Lowry & Nelson, 1985*; *Frost & Lowry, 1995*) were used to estimate availability probability (S4).

Unless otherwise stated, the following text uses unbolded symbols to denote scalars, lower case bolded symbols to denote vectors, and upper case bolded symbols to denote matrices. See S1 for a *Glossary of Notation and Abbreviations*.

## Design-based estimator

Although our primary focus is on developing density surface models for EBS belugas, previous abundance estimates for this stock were generated using a design-based estimator. A basic Horvitz-Thompson-like line-transect estimator of animal density is (*Buckland et al., 2001*; *Burt et al., 2014*):

$$\hat{D} = \frac{1}{a} \sum_{j=1}^{n_g} \frac{S_j}{\hat{p}(\mathbf{z}_j; \hat{\boldsymbol{\theta}})} \tag{1}$$

where

| | |
|---|---|
| $n_g$ | : number of groups detected; |
| $S_j$ | : size of group indexed by $j$; |
| $a$ | : area searched during line-transect survey, where $a = 2Lw$, $L$ is the total length of transects surveyed, and $w$ is the width of the strip searched on one side of the aircraft; |
| $\hat{p}(\mathbf{z}_j; \hat{\boldsymbol{\theta}})$ | : model-based estimate of the overall probability that an observer detects group $j$, given covariates $\mathbf{z}_j$ that affect detectability. This term accounts for all sources of perception and availability bias (*Marsh & Sinclair, 1989*; S4); |
| $\hat{\theta}$ | : parameter estimates required to estimate detection probabilities. |

To derive an estimate of the total number of animals in the study area ($\hat{N}$), we multiply the total study area size, $A$, by the density estimate from Eq. (1):

$$\hat{N} = A\hat{D}. \tag{2}$$

Under this formulation, inference proceeds by first fitting detection function models to observed distances and other covariates to produce estimates of detection parameters (*i.e.*, $\hat{\theta}$), before applying Eq. (1) in a second step. The abundance estimator in Eq. (2) is unbiased if certain assumptions about the survey design and realization hold (*Buckland et al., 2001*; *Hedley & Bravington, 2014*). Hence, this $\hat{N}$ is referred to as a design-based estimator.

In addition to allowing calculation of the design-based estimator, we also use these estimates of detection probability when fitting DSMs. In the following, we shorten notation such that $p_j = \hat{p}(\mathbf{z}_j; \hat{\boldsymbol{\theta}})$ for sightings and $p_i = \hat{p}(\mathbf{z}_i; \hat{\boldsymbol{\theta}})$ for segments. For more information about detection probability calculations, see S4.

## DSMs: marginal likelihood

As with most DSM implementations, we construct a spatial model for counts of individuals, which in our case were summarized over 10-km transect segments (see *Eastern Bering Sea beluga case study* for further information). For all DSMs, we write a generic marginal likelihood of a parameter vector, $\boldsymbol{\xi}$, given observed counts of individual animals, $\mathbf{c}$, and other known variables, $\mathbf{x}$, as

$$\mathcal{L}(\boldsymbol{\xi}; \mathbf{c}, \mathbf{x}) = \int_{\boldsymbol{\eta}} [\mathbf{c}|\boldsymbol{\xi}, \boldsymbol{\eta}, \mathbf{x}][\boldsymbol{\eta}|\mathbf{x}, \boldsymbol{\xi}] d\boldsymbol{\eta}. \tag{3}$$

Here, $[\mathbf{c}|\xi, \boldsymbol{\eta}, \mathbf{x}]$ is the conditional probability density function of observed counts, given parameters, random effects ($\boldsymbol{\eta}$), and known covariates. The counts represent the

number of animals detected on rectangular transect segments (with width $2w$, as in the design-based estimator). The component $[\boldsymbol{\eta}|\mathbf{x}, \boldsymbol{\xi}]$ represents the distribution of random effects. We use the integral to indicate that the random effects will be integrated out of the joint likelihood—in our case using the Laplace approximation available in TMB software (*Kristensen et al., 2015*). As usual in likelihood-based inferential statistics, the likelihood is viewed as a function of the unknown parameters, $\boldsymbol{\xi}$.

In order to derive $[\mathbf{c}|\boldsymbol{\xi}, \boldsymbol{\eta}, \mathbf{x}]$, we must first specify a suitable probability mass or density function. Although it is customary to specify probability mass functions for count data, initial exploration of Poisson and negative binomial distributions indicated considerable lack-of-fit when applied to our EBS beluga data set. Specifically, model diagnostic plots examining the relationship between the mean and variance in the residuals compared to the theoretical distribution (*Ver Hoef & Boveng, 2007*), and quantile residuals computed using a probability integral transform (PIT; *Dunn & Smyth, 1996*) and visualized using the R package DHARMa (*Hartig, 2022*), showed that Tweedie distributions (*Jørgensen, 1987*; *Dunn & Smyth, 2005*; *Kendal, 2004*) provided a better fit to the data. Therefore, we adopted a parameterization based on the Tweedie distribution.

The Tweedie distribution provides increased flexibility compared to the Poisson and negative binomial distributions, allowing a diversity of shapes and accommodating zero-inflation. It is a specific case of an exponential dispersion model, with mean $\mu$, and variance $V(\mu) = \phi\mu^\rho$ (*Dunn & Smyth, 2005*). We specifically set the range of $\rho$ to be on $1 < \rho < 2$, a parameterization variously known as "compound Poisson", "compound gamma", or "Poisson-gamma" (*Dunn & Smyth, 2005*; *Kendal, 2004*). This distribution has support on the non-negative real line, although authors often use this distribution for non-negative integers (*e.g.*, counts; *Kendal, 2002*; *Miller et al., 2013*; *Sigourney et al., 2020*), which is our approach in this paper. *Kendal (2002)* and *Kendal (2004)* discuss the relationship between the Tweedie distribution and Taylor's power law in ecology, which explains clustered spatial distributions as manifestations of power function relationships between the variance and mean number of organisms in an area (*Taylor, 1961*).

For $1 < \rho < 2$, the Tweedie distribution does not have a closed form, but can be evaluated numerically (*e.g.*, using the 'dtweedie' function in the TMB library). Therefore, we symbolically write

$$[\mathbf{c}|\boldsymbol{\xi}, \boldsymbol{\eta}, \mathbf{x}] \equiv \prod_i \text{Tweedie}(c_i; \mu_i, \phi, \rho). \tag{4}$$

That is, the joint likelihood of observed counts on transect segments indexed by $i$ is a product of conditionally independent univariate Tweedie density functions, with mean $\mu_i$ and constant dispersion and power parameters, $\phi$ and $\rho$. The mean, $\mu_i$, is a function of fixed and random effects, and "known" detection probability and survey coverage offsets, such that

$$\boldsymbol{\mu} = \exp(\beta_0 + \boldsymbol{\delta} + \log(\mathbf{a}) + \log(\mathbf{p})). \tag{5}$$

Here, $\beta_0$ represents an intercept parameter (no other fixed effects were included in our models), $\boldsymbol{\delta}$ is a vector of 'realized' random effects for transect counts, $\mathbf{a}$ is a vector of the area

surveyed for each transect segment ($a_i = 2L_i w$), and **p** is a vector of the overall detection probability (including both availability and perception bias corrections) for each segment ($p_i$, see S4 Eq. 6).

Note that this parameterization requires that any covariates used to estimate detectability relate only to the transect segment; observation-specific covariates (*e.g.*, color, group size) cannot be used in this parameterization. See *Miller et al. (2013)* for an alternative parameterization that allows observation-specific covariates by specifying the response variable to be an estimate of bias-corrected abundance.

The actual dimension of random effects often differs from the number of transect segments. Specifically, we model $\boldsymbol{\delta} = \mathbf{A}\boldsymbol{\eta}$, where the matrix **A** has dimension $(n_i, n_\eta)$, with $n_\eta$ denoting the true number of random effects, and $n_i$ the number of transect segments. Next, we elaborate on the random effects specifications.

## Random effects specifications

We have yet to describe $[\boldsymbol{\eta}|\mathbf{x}, \boldsymbol{\xi}]$ in Eq. (3). This component defines the specification of spatially autocorrelated random effects. For a given survey period (year) the data likelihood Eq. (4) is the same for all the models that we considered, so the random effects specification is the only difference among the DSMs we developed.

For all models, random effects were assumed to be drawn from a multivariate normal distribuion with mean zero, and a spatially patterned covariance matrix, $\boldsymbol{\Sigma}$:

$$\boldsymbol{\eta} \sim \text{Multivariate normal}(\mathbf{0}, \boldsymbol{\Sigma}).$$

Spatial autocorrelation is imparted by constraints on the $(n_\eta \times n_\eta)$ $\boldsymbol{\Sigma}$ matrix. In practice, we chose to work with a precision matrix $\mathbf{Q} = \boldsymbol{\Sigma}^{-1}$, which was often sparse, enabling greater computational efficiency.

We employed two related, but conceptually different types of models to specify **Q**: stochastic partial differential equations (SPDEs) to approximate Matérn geostatistical models (*Lindgren, Rue & Lindström, 2011*), and spline-based models. The latter are commonly used in generalized additive models (*e.g.*, *Wood, 2006*), where smooth terms are often viewed as penalized fixed effects. However, it is also possible to conceptualize smooths as mean-zero random effects, with an associated precision matrix (*Miller, Glennie & Seaton, 2020*), which is the approach that we used in this paper. Further details on how **Q** is specified for individual models is provided below.

We conducted all analyses in the R programming environment (version 4.3.2; *R Core Team, 2023*) using the TMB package (*Kristensen et al., 2015*) to formulate marginal log-likelihoods and generate DSM parameter estimates, the mgcv package (*Wood, 2006*) to set up spatial spline bases, and R-INLA (*Rue, Martino & Chopin, 2009*) to create Delaunay triangulation meshes for Stochastic Partial Differential Equation (SPDE) models. All data and code used in this paper have been uploaded to a public GitHub repository at https://github.com/meganferg/FergusonEtal_20250125_EBS_Beluga_DSM and are available on Zenodo at https://doi.org/10.5281/zenodo.16269135.

### Matérn covariance models

*SPDE Matérn model.* We used an SPDE approximation to model the precision matrix associated with the Matérn covariance function (*Lindgren, Rue & Lindström, 2011*). This required establishing a set of $n_\eta$ vertices at locations $\mathbf{s} \subset \mathbb{R}^2$ (often termed "knots"). The knot locations $\mathbf{s}$ are each associated with one of the $n_\eta$ random effects. We used the function "inla.mesh.2d" from R-INLA to specify knot locations, creating a triangular mesh (*i.e.,* spatial basis) that allows animal density to be predicted at any location on the mesh.

A particular mesh is defined by a number of characteristics, including the spatial domain (*i.e.,* geographic boundary), location of knots, and maximum and minimum distances between knots. Additionally, boundary conditions must be imposed to create SPDE solutions on bounded domains (*Lindgren & Rue, 2015*). R-INLA uses Neumann conditions, which results in variance inflation by a factor of two along straight boundaries and a factor of four near right-angled corners. At a distance equal to the geostatistical range (*i.e.,* the distance at which the correlation between two points approaches zero), the boundary effect is negligible. To eliminate boundary effects in the area of interest, *Lindgren & Rue (2015)* recommend extending the outer boundary of the spatial domain by a distance at least equal to the range. Knot density can be reduced in the outer buffer area to minimize the additional computational burden of the knots located far from the data.

Different meshes could result in different estimates of animal density. There are no strict rules on how to create a mesh for a particular dataset. Therefore, we used preliminary analyses with a variety of meshes to guide our decisions on which mesh was best suited to each year's data. For example, a certain mesh might be a poor fit to the data because the numerical optimization of the geostatistical range parameter can fail if spatial autocorrelation occurs at a much finer scale than the minimum distance between knots. In general, we followed the suggestions of *Belmont (2022)* to create the meshes. Initial knot locations were placed at the transect segment midpoints. The maximum distance between knots in the buffer area was twice that in the aerial survey boundaries. The minimum distance between knots equaled $\frac{1}{5}$ the maximum distance between knots. The extension radius used to set the overall boundary of the spatial domain (hence, the width of the buffer area) was approximately 35% of the diameter of the aerial survey study area.

Interpolations of random effects between knot locations and data locations are made with a bilinear interpolation matrix ($\mathbf{A}$), where the data location is taken to be the centroid of each 10-km transect segment. We used the R-INLA function "inla.spde.make.A" to create interpolation matrices. Interpolation matrices are completely determined by the underlying mesh and the data locations, and are nonzero for only three elements of each row (corresponding to the three triangular vertices that surround a given point).

To define a precision matrix for the Matérn covariance model at knot locations, we used the function "inla.spde2.matern" in R-INLA (*Rue, Martino & Chopin, 2009*; *Lindgren, Rue & Lindström, 2011*), which generates three structure matrices, $M_0$, $M_1$, and $M_2$. The precision matrix is then specified using these three matrices, together with two unknown parameters, $\tau$ and $\kappa$:

$$\mathbf{Q} = \tau^2 (\kappa^4 \mathbf{M}_0 + 2\kappa^2 \mathbf{M}_1 + \mathbf{M}_2). \tag{6}$$

Here, $\tau$ can be be interpreted as a precision parameter and $\kappa$ as an inverse geostatistical range parameter. Equation (6) results from applying finite-element methods to approximate a stochastic partial differential equation representing diffusion. Using notation from *Lindgren, Rue & Lindström (2011)*, we see that a diffusive SPDE for second-order adjacency results in a precision with the form:

$$\mathbf{Q} = \mathbf{K}\mathbf{C}^{-1}\mathbf{K} \tag{7}$$

where $\mathbf{K} = \kappa^2(C + G)$ is interpreted as the instantaneous diffusion rate, and where $\mathbf{C}$ and $\mathbf{G}$ are both sparse matrices. Replacing $\mathbf{C}$ with a diagonal matrix $\tilde{\mathbf{C}}$ (so that $\tilde{\mathbf{C}}^{-1}$ remains sparse), plugging in $\mathbf{K}$, and simplifying, we then obtain Eq. (6) where $\mathbf{M_0} = \tilde{\mathbf{C}}$, $\mathbf{M_1} = \mathbf{G}$, and $\mathbf{M_2} = \mathbf{G}\tilde{\mathbf{C}}^{-1}\mathbf{G}$. However, we retain M-notation in Eq. (6) to maintain consistency with terminology that is common when using the R-INLA package.

Ultimately, the complexity (*i.e.*, effective degrees of freedom) of the SPDE Matérn model is a function of the mesh resolution (*i.e.*, the number of knots in the mesh) and the maximum likelihood estimates of the correlation function parameters, $\kappa$ and $\tau$. Asymptotically, increasing the number of knots in the mesh will converge on the "best" functional approximation of the underlying function, and further increases in mesh resolution will not influence model results. However, this resolution is rarely achieved in practice due to computational limitations. Care must be taken to construct an appropriate mesh and specify SPDE parameters (*e.g.*, boundary conditions, anisotropy) to approximate the correlation function based on knowledge of the system, followed by implementation of model evaluation tools to identify an appropriate mesh. The primary determinant of SPDE Matérn model complexity (and, therefore, conditional deviance explained) are the MLEs of $\kappa$ and $\tau$, which are informed by the data. Maximum likelihood estimation of these correlation function parameters has good asymptotic properties (*Thorson & Kristensen, 2024*).

*SPDE Matérn model with barriers.* The SPDE model defined above approximates a stationary, isotropic Matérn covariance function. Conditional on $\tau$ and $\kappa$, the only variable affecting spatial autocorrelation is the distance between knots. However, in areas with complex coastlines (such as islands, bays, peninsulas, and points), it is plausible that spatial connectivity for the distribution and density of marine animals would be interrupted by land barriers, making points that are close together in the contiguous ocean more "alike" than two points separated at the same distance on the opposite sides of a land barrier. Therefore, we implemented an alternative SPDE Matérn model that accounts for land-based barriers. Specifically, we followed the approach outlined by *Bakka et al. (2019)*, where locations that occur on land are assigned a small, fixed effective range value, and the range parameter for locations at sea is estimated during model fitting. To implement the $\mathbf{Q}$ matrix for this model in TMB, we used the code template at https://github.com/skaug/tmb-case-studies/tree/master/spdeBarrier. This essentially specifies a high value for decorrelation rate $\kappa$ for knots over land, to ensure that correlations between locations in water are calculated from the set of paths over water.

### Spline-based smoothers

We considered several types of spline-based smoothers as alternatives for specifying spatial random effects. In each case, we used mgcv to construct spline bases and appropriate penalization matrices, then passed these into TMB when formulating our marginal log-likelihood. We implemented three types of bivariate smoothing splines: isotropic, thin plate regression splines (tprs) with shrinkage (*Wood, 2003*); anisotropic tensor product splines (*Wood, 2017*) comprising tprs with shrinkage; and isotropic soap film smoothing splines (*Wood, Bravington & Hedley, 2008*). The first two types of splines treat spatial correlation as depending on distance only (isotropic tprs), or distance and direction (tensor product splines). The soap film smoother allows spatial correlation to be interrupted when there are barriers, such as land, between suitable habitat. We reasoned this would be a desirable property given the complex coastline in our study area, which included multiple peninsulas and estuaries (Fig. 1).

*Bivariate and isotropic thin plate regression spline.* For the bivariate and isotropic thin plate regression spline, we used the gam() function in the mgcv R package to construct a bivariate "ts" spline basis of easting and northing for observed data ($\mathbf{A}$, typically referred to as a design matrix in this context), an interpolation matrix for predictions ($\mathbf{A}^{pred}$), and a penalization matrix, $\mathbf{S}$. We then set $\mathbf{Q} = \lambda \mathbf{S}$ in our TMB optimization, where the smoothing parameter $\lambda$ was treated as an estimated parameter. This procedure follows the example by H. Skaug at https://github.com/skaug/tmb-case-studies/tree/master/pSplines.

*Tensor product smoother.* The tensor product smoother produces an anisotropic spline basis, allowing different correlations on the dimensions corresponding to easting and northing in our analysis. For the tensor product smoother, we again used a gam() function in the mgcv R package to construct a "ts" spline basis for observed data ($\mathbf{A}$) and an interpolation matrix for predictions ($\mathbf{A}^{pred}$). In this case, mgcv produces two penalization matrices, $\mathbf{S_1}$ and $\mathbf{S_2}$ (one for easting and one for northing). Following code from D. Miller (https://github.com/dill/mgcvminusminus), we set $\mathbf{Q} = \lambda_1 \mathbf{S_1} + \lambda_2 \mathbf{S_2}$, where $\lambda_1$ and $\lambda_2$ are treated as estimated parameters.

*Soap film smoother.* The soap film smoother (*Wood, Bravington & Hedley, 2008*; *Miller & Wood, 2014*) is another approach to constructing a smooth surface over space where correlation does not persist over boundaries (*e.g.*, penninsulas). To produce $\mathbf{A}$, $\mathbf{A}^{pred}$, and penalization matrices, we again used mgcv. In particular, we supplied the gam() function with a dataframe delineating study area boundaries. Like the tensor product smoother, the soap film smoother option in mgcv produces two penalization matrices, $\mathbf{S_1}$ and $\mathbf{S_2}$, this time associated with boundaries and internal space, respectively. However, we constructed the precision matrix in the same manner (*i.e.*, $\mathbf{Q} = \lambda_1 \mathbf{S_1} + \lambda_2 \mathbf{S_2}$, where $\lambda_1$ and $\lambda_2$ are treated as estimated parameters).

## Prediction

For each model and year of analysis, we summed predictions of the number of belugas in each hexagonal grid cell $h$ in our study area (see *Eastern Bering Sea beluga case study* for

how these were defined), applying epsilon bias-correction to this total (see *Correcting for detransformation bias*). Specifically, we calculated

$$\hat{N}_h = \exp(\hat{\beta}_0 + \hat{\delta}_h + \log(a_h)). \tag{8}$$

Note that $a_h$ gives the area of ocean in hexagon $h$ (*i.e.,* omitting land). There is no offset for detection probability because we are interested in all belugas, not just those that are detectable and detected. The vector of "realized" random effects $\hat{\boldsymbol{\delta}}$ is calculated as $\hat{\boldsymbol{\delta}} = \mathbf{A}^{pred}\hat{\boldsymbol{\eta}}$ where $\hat{\boldsymbol{\eta}}$ is the value of $\boldsymbol{\eta}$ that maximizes the joint likelihood conditional upon the maximum likelihood estimator (MLE) $\hat{\boldsymbol{\xi}}$ for fixed effects (termed the empirical Bayes estimator for $\boldsymbol{\eta}$). This predictor for $\hat{N}_h$ is called the "plug-in estimator" because it plugs in the empirical Bayes estimator as if it were fixed. For SPDE models, the $(n_h, n_\eta)$ interpolation matrix, $\mathbf{A}^{pred}$ was constructed using the "inla.spde.make.A" function in R-INLA, using the centroid of each hexagon as the prediction location. For spline-based smoothers, we obtained $\mathbf{A}^{pred}$ using mgcv's "predict()" function with "type=lpmatrix", again using the centroids of each hexagon as prediction locations. An estimate of total abundance is then calculated as $\hat{N} = \sum_h \hat{N}_h$. The epsilon bias-correction was applied to $\hat{N}$.

## Model evaluation and final candidate model selection

To evaluate DSMs and select the final candidate DSMs for the ensemble model, we advocate using several criteria, including: examining conditional PIT residuals *via* the DHARMa package (*Hartig, 2022*); extrapolation metrics (defined below); and visual examination of maps showing the DSM predictions overlaid with the sightings used to build the models. We provide a detailed example in the *Eastern Bering Sea beluga case study*.

To identify models whose predictions might be unreliable due to extrapolation bias, we considered two types of ad hoc metrics. First, for each combination of model and cell (*i.e.,* hexagon, $h$), we computed the following ratio:

$$\frac{\lambda_{m,h}}{\lambda_{m,max}}. \tag{9}$$

Here $\lambda_{m,h}$ is the predicted abundance from model $m$ for unsampled location $h$, and $\lambda_{m,max}$ is the maximum predicted abundance across all sampled cells. Second, for each model we counted the number of unsampled cells (*i.e.,* cells that did not have line-transect survey effort) with predicted abundance exceeding the maximum predicted abundance in sampled cells. These procedures are motivated by a generalized version of Cook's independent variable hull (*Cook, 1979*; *Conn, Johnson & Boveng, 2015*).

## Uncertainty estimation
### Correcting for detransformation bias

Random effects $\boldsymbol{\eta}$ are treated as random variables (and marginalized across) during maximum likelihood estimation, but are then treated as if they were fixed at the modes of their distributions (conditional on the MLEs for the fixed effects) by the plug-in estimator. However, $\boldsymbol{\eta}$ will generally have substantial variance and skewness, and this will cause the plug-in estimator to be a poor estimator for the expectation of $N$ when integrating across the distribution for random effects.

To better estimate the expectation for $N$, we employed the epsilon bias-correction procedure described by *Thorson & Kristensen (2016)* and implemented in TMB to obtain estimates. This epsilon method corrects for both the nonlinearity of the transformation (*i.e.,* exponentiation in Eq. (8)) and the variance and skewness of random effects. *Thorson & Kristensen (2024)* Chap. 6 shows a closed-form calculation for the epsilon method in a simplified scenario involving a single (scalar-valued) random effect, and confirms that it provides very close to the known expectation when transforming skewed random variables with a number of different nonlinear functions.

### Variance estimation

We relied on the law of total variance to construct an unconditional variance estimator for each DSM that includes uncertainty from the MCDS contribution to detection probability, $\mathbf{p}_g$ (S5). Specifically, we calculated

$$\hat{\mathrm{Var}}(\hat{N}) = \mathbb{E}(\hat{\mathrm{Var}}(\hat{N}|\tilde{\mathbf{p}}_g)) + \hat{\mathrm{Var}}(\mathbb{E}(\hat{N}|\tilde{\mathbf{p}}_g)). \qquad (10)$$

The first part of Eq. (10), $\mathbb{E}(\hat{\mathrm{Var}}(\hat{N}|\tilde{\mathbf{p}}_g))$, is the expected variance of the abundance estimator given a particular realization of detection probability, $\tilde{\mathbf{p}}_g$. We approximated this component with $\hat{\mathrm{Var}}(\hat{N}|\hat{\mathbf{p}}_g)$, which is the variance of the plug-in abundance estimator conditioned on the MLEs for detection probability from the MCDS analysis. This variance estimate is produced by the TMB software, using the algorithm detailed below (see *Conditional variance of abundance estimator*).

The second component of Eq. (10), $\hat{\mathrm{Var}}(\mathbb{E}(\hat{N}|\tilde{\mathbf{p}}_g))$, in effect gives the variance of the mean, representing how estimates of abundance vary depending on the values of $\mathbf{p}_g$ that are sampled. To approximate $\hat{\mathrm{Var}}(\mathbb{E}(\hat{N}|\tilde{\mathbf{p}}_g))$, we used the following bootstrap procedure (see S5 for pseudocode):

1. For $k \in 1, 2, \ldots, K$, sample $\mathbf{p_g}^{(k)} \sim f(\mathbf{p}_g)$, where $f(\mathbf{p}_g)$ is the joint predictive distribution of detection probabilities from the MCDS detection function analysis. In practice, each sample $\mathbf{p}_g^{(k)}$ was obtained by assuming that the parameters of the detection function had a multivariate normal distribution on the logit scale.

2. For each $k$, fit a TMB DSM to the beluga data, treating $\mathbf{p}_g = \mathbf{p}_g^{(k)}$ as a fixed value, and record the abundance estimate, $\hat{N}^{(k)}$. To approximate $\hat{\mathrm{Var}}(\mathbb{E}(\hat{N}|\tilde{\mathbf{p}}_g))$ for the epsilon bias-corrected abundance estimate of $\hat{N}$, the value $\hat{N}^{(k)}$ should be the bias-corrected estimate, not the plug-in estimate.

3. Approximate $\hat{\mathrm{Var}}(\mathbb{E}(\hat{N}|\tilde{\mathbf{p}}_g))$ as $K^{-1}\sum_k (\hat{N}^{(k)} - \bar{N})^2$, where $\bar{N}$ is the mean abundance estimate from all $K$ bootstrap iterations.

Following application of this procedure, to generate estimates of total uncertainty in the abundance estimate from each individual DSM (*i.e.,* $CV_{tot}(\hat{N}_m)$ in Eq. 5 of S5), the delta method (*Dorfman, 1938*) could be used to incorporate the uncertainty due to independent estimates of transect detection probability or availability probability.

*Conditional variance of abundance estimator.* We compute an estimator for the variance of Eq. (8) that accounts for uncertainty in both fixed and random effects. We call this a conditional estimator because we are specifically conditioning on a fixed vector of detection

probabilities. Although different estimators are available, TMB software uses the estimator from *Kass & Steffey (1989)*. This involves calculating the joint precision $\mathbf{Q}_{joint}$ for fixed and random effects:

$$\mathbf{Q}_{joint} = \begin{pmatrix} \mathbf{H_1} & -\mathbf{H_1}\nabla \\ -\nabla^{\mathbf{t}}\mathbf{H_1} & \nabla^{\mathbf{t}}\mathbf{H_1}\nabla + \mathbf{H_2} \end{pmatrix} \tag{11}$$

where $\mathbf{H}_2$ is the matrix of second derivatives for $\log\mathcal{L}(\boldsymbol{\xi}|\mathbf{c},\mathbf{x})$ (the "outer Hessian matrix"), $\mathbf{H}_1$ is the matrix of second derivatives for $\log([\mathbf{c}|\boldsymbol{\xi},\boldsymbol{\eta},\mathbf{x}][\boldsymbol{\eta}|\mathbf{x},\boldsymbol{\xi}])$ conditional upon the MLE for fixed effects $\boldsymbol{\xi}$ (the "inner Hessian matrix"), and $\nabla$ is the matrix of gradients of predicted random effects with respect to fixed effects (the "outer Jacobian matrix").

We then compute the variance for derived quantity $\hat{N}$ from this joint precision. We specifically calculate the gradient $\mathbf{J}$ of $\hat{N}$ with respect to the vector of fixed and random effects. We then compute

$$\hat{\mathrm{Var}}(\hat{N}|\hat{\mathbf{p}}_g) = \mathbf{J}\mathbf{Q}_{joint}^{-1}\mathbf{J}^t. \tag{12}$$

## Ensemble model

Fitting multiple DSMs to sightings raises the question of which model, or collection of models, should be used to generate a final abundance estimate and density surface. The question is particularly important when different models produce markedly different estimates of abundance. We chose to base ultimate inference on an ensemble (*Araújo & New, 2007*), whereby estimates from different models are averaged to produce a final estimate. Specifically, we compute

$$\hat{N}_{ens} = \sum_m w_m \hat{N}_m \tag{13}$$

where $\hat{N}_m$ is the MLE of abundance from TMB for each model $m$. The model weight is $w_m$, where $\sum_m w_m = 1.0$. The advantage of averaging models is that there is often a reduction in prediction error (*Burnham & Anderson, 2002*; *Dormann et al., 2018*).

There are different approaches for setting the model weights (*Dormann et al., 2018*). For instance, a common approach is to use Akaike's information criterion (AIC) associated with fitted models to calculate weights (*Burnham & Anderson, 2002*). However, calculation of AIC weights relies on the complexity of a model, often computed as the effective degrees of freedom from a generalization of the hat-matrix, and this is difficult to compute in a hierarchical model using maximum-likelihood methods. Instead, we used equal model weights, which have been shown to perform well in prediction of species distributions (*Dormann et al., 2018*). This procedure has the added advantage that a single model with an extremely high or low abundance estimate will not dominate inference.

The variance of model-averaged predictions was calculated using the standard unconditional variance estimator (*i.e.*, *Burnham & Anderson, 2002*):

$$\hat{Var}(\hat{N}_{ens}) = \left[\sum_m w_m \sqrt{Var(\hat{N}_m) + (\hat{N}_m - \hat{N}_{ens})^2}\right]^2. \tag{14}$$

## Eastern Bering Sea beluga case study

The EBS beluga survey design and data collection methods are similar to those previously described in *Ferguson, Conn & Thorson (2024)*; the analytical methods and results reflect improvements to *Ferguson, Conn & Thorson (2024)*. The data collection methods and sighting and effort summaries are presented in S2 for the aerial line-transect surveys and S3 for the aerial imagery. All aerial surveys were approved by the Alaska Fisheries Science Center/Northwest Fisheries Science Center Institutional Animal Care and Use Committee (IACUC Number NW/AK2016-6). The analytical methods used to estimate detection probabilities and corresponding results are presented in S4. See *Frost, Lowry & Nelson (1985)* and *Frost & Lowry (1995)* for details about the VHF telemetry data and analyses. There were no estimates of uncertainty for availability probability (*Ferguson et al., 2023*); therefore, this parameter was included as a known constant in the offset for the DSM Eq. (5).

To derive spatially-explicit estimates of EBS beluga abundance, we constructed density surface models separately for each year, 2017 and 2022. The 2017 study area boundary corresponded to the area covered by the geographic strata in *Lowry et al. (2017)* (Fig. 1). The ABWC advocated for a larger survey area in 2022. They noted that Indigenous knowledge has confirmed that the southern extent of the EBS beluga stock's distribution during early summer extends farther south than the historical survey boundaries. Therefore, the 2022 survey area extended south of Hooper Bay (Fig. 1). The study area boundary used to estimate abundance in 2022 excluded Scammon Bay because that bay was not included in the 2022 survey design, there was no survey effort inside the bay (*i.e.,* east of the barrier islands), and we have insufficient information about beluga habitat to make inference about beluga density in Scammon Bay based on density in the area surveyed outside Scammon Bay.

DSMs were constructed using aerial line-transect sighting and effort summaries for 10-km segments of transect effort. This segment length is approximately the distance between adjacent transects (9.3 km). The segments were created by sequentially slicing transect effort conducted in Beaufort Sea State ≤4, beginning with the start of each transect. End segments <10 km were added to adjacent segments so that all segments used in the analysis were ≥10 km. Predictions from the DSM were based on a hexagonal grid with cell midpoints located 10 km apart. All geospatial data were projected into an equidistant conic projection (false easting: 0.0; false northing: 0.0; central meridian: −164.0°; latitude of origin: 63.5°; standard parallels: 62.5°, 64.5°; WGS84 datum; linear unit: kilometer), and this projection was used when calculating cell areas and distances for the spatial correlation functions or splines.

The DSMs required segment-specific estimates of detection probability, $p_i$ Eq. (5). The best-fitting MCDS detection function for EBS belugas included covariates for Beaufort Sea State (integer-valued) and turbidity (binary) (S4). To build the DSMs, effort data for these variables were summarized by segment. The segment-specific Beaufort Sea State variable was calculated as the average value of integer-valued Beaufort Sea State for all records that were located on the segment; all records were weighted equally. The segment-specific turbidity variable was calculated by assigning the binary turbidity variable an integer value (no = 0; yes = 1), computing the average of the integer-valued turbidity values for all records located on the segment, and rounding the result. For example, if segment $i$

comprised three data records with turbidity "yes", "yes", and "no", the average of their integer-valued analogs would be $1 + 1 + 0 = 0.67$, which rounds to 1, so the segment would be designated as turbidity = "yes".

For 2017 and 2022, 13 and 17 DSMs, respectively, were constructed and examined. For each year, we allowed at most one Matérn and one SPDE Matérn model with barriers to be included in the ensemble. Overviews of key aspects of each model are provided in Tables S6.1 and S6.2. To ensure that all models for a given year were allowed the same flexibility to construct the spatial surface, we specified the DSMs for each year using approximately the same number of random effects. For each year, the number of random effects for all DSMs was based on the number of random effects (*i.e.,* vertices) in that year's SPDE Matérn model, which was determined using the guidelines provided in *Belmont (2022)*. The knot locations for the soap film smoothers corresponded to the locations of the vertices for the corresponding SPDE Matérn model that were within the soap boundary. This resulted in fewer total knots for the soap film smoother than the corresponding SPDE Matérn model; however, both models had the same number of knots inside the soap boundary, and the soap film smoother was specified to have the same number of random effects as the corresponding SPDE Matérn model. The total number of random effects for the bivariate tensor product smoother corresponds to the product of each of the basis dimensions used to build the smoother, and it was not possible to exactly match the number of random effects in the corresponding SPDE Matérn model.

Identical DSMs were fit in mgcv and TMB, with the exception of the barrier SPDE models, for which mgcv functions defining this type of basis were not available, so they were constructed only in TMB. We constructed identical models in both software platforms for two reasons: (1) to apply the methods presented in *Miller, Glennie & Seaton (2020)* to the EBS beluga data and confirm that nearly identical results could be derived using mgcv and TMB; and (2) to evaluate whether the existing "ds_varprop" function in the dsm package (*Bravington, Miller & Hedley, 2021*; *Miller et al., 2022*) would be an alternative to Eq. (10) and the methods presented in S5 for propagating uncertainty from the MCDS detection function model into the overall estimate of uncertainty in the abundance estimate. However, as of dsm version 2.3.3, the "dsm_varprop" function could not propagate errors through SPDE models (M Ferguson with D Miller, pers. comm., 2023).

Because the variance in the MLE of abundance from TMB for each candidate DSM, $Var(\hat{N}_m)$, did not include uncertainty from the estimate of transect detection probability in the beluga case study (*i.e.,* $\hat{p}_{MR}(0, \mathbf{z}_j; \hat{\boldsymbol{\theta}}_{MR})$ in S4), we applied the delta method to add this component of uncertainty to $\hat{Var}(\hat{N}_{ens})$, resulting in $\hat{Var}_{tot}(\hat{N}_{ens})$ (see Eq. 4 of S5).

For comparison with the 2017 EBS beluga DSM results, we examined the conventional abundance estimate derived using the post-stratified design-based abundance estimator of *Ferguson et al. (2023)*. Their variance estimator had three components: (1) variation from uncertainty in estimating the MCDS parameters; (2) variation from uncertainty in estimating transect detection probability; and (3) variation in abundance due to random sample selection. Two conventional abundance estimates for 2022 were derived using the post-stratified design-based estimator. One conventional estimate for 2022 used the same strata as the conventional abundance estimate for 2017 (*i.e.,* the *Lowry et al. (2017)* strata;

*Ferguson et al. (2023)*). The other conventional estimate for 2022 used geographic strata covering the full extent of the 2022 study area (Fig. 1). The modified geographic strata for 2022 incorporated the latitudinal boundaries from *Lowry et al. (2017)* (Figs. 1, S2.2). Unlike the *Lowry et al. (2017)* strata, the modified 2022 strata extended to the western border of the 2022 study area and two new strata were created to encompass the northernmost and southernmost portions of the 2022 study area. Both conventional abundance estimates for 2022 used the methods of *Ferguson et al. (2023)*, with a MCDS detection function model based on the pooled data from 2017 and 2022 (discussed briefly above and in S4).

The detection function model used in the post-stratified design-based estimator for 2017 was based on only a single year of data (*Ferguson et al., 2023*), whereas the detection function model used in our model-based estimators for 2017 and 2022 was based on data from both years pooled (S4). However, the CV of the former detection function model was 0.043 and the CV of the latter was 0.037, only trivially smaller; therefore, we do not believe that this difference in detection function models affected our overall comparison of the precision in the different abundance estimators.

## RESULTS

Here, we focus on results of the EBS beluga case study. None of the 13 candidate models for 2017 exhibited signs of extrapolation bias based our extrapolation diagnostics (Table S6.1). For 2022, three of the 17 candidate models had one cell (out of a total of 554 cells in the prediction grid) with extrapolation ratios $> 1.0$ Eq. (9), and all three of those models were SPDE Matérn models with barriers (Table S6.2). Based on these metrics, we did not find evidence for concern about extrapolation bias.

The number of models per year that were selected for inclusion in the ensemble model average was not chosen a priori. Rather, we examined PIT residuals *via* the R package DHARMa (*Hartig, 2022*; S6), extrapolation metrics, and visual inspection of maps of $\hat{N}$ predictions, sightings, and effort to narrow the field to four candidate models for 2017 and three candidate models for 2022 (Tables 1; S6). For both years, the SPDE Matérn models with 60-km maximum edge length were included in the ensemble. The SPDE Matérn model with barriers was eliminated from the ensemble model for 2017 because the residual analyses from the DHARMa package showed quantile deviations and the combined adjusted quantile tests were significant. The tensor product smoother was eliminated from the ensemble model for 2022 because it had relatively low percent deviance explained, even though it had 323 random effects compared to 308 for the corresponding SPDE Matérn model, and models in 277 out of the 500 iterations failed to converge in the parametric bootstrap. The bivariate isotropic tprs was eliminated from the 2022 ensemble because residual analyses from the DHARMa package showed significant quantile deviations and the estimated values of the Tweedie dispersion and power parameters were suspiciously low relative to all other models.

The candidate DSMs included in the ensemble for 2017 were the SPDE Matérn, soap film smoother, tensor product smoother, and bivariate isotropic thin plate regression spline (Fig. 2). The candidate DSMs included in the ensemble for 2022 were the SPDE

**Table 1  Estimated abundance ($\hat{N}$) of Eastern Bering Sea belugas from models fitted to 2017 and 2022 aerial line-transect survey data.** For spatial models, we present both uncorrected estimates and those that employed epsilon bias correction ("Corrected"). Because precision estimates were numerically intensive to calculate, and only of primary interest for epsilon bias-corrected models, we provide estimated CVs (parentheses) for epsilon bias-corrected estimates and for the full area surveyed each year. We also provide point estimates of abundance based on the 2022 survey data that were restricted to the area within the 2017 strata to allow comparison between years for the same region. SPDE = SPDE Matérn model. SPDE with barriers = SPDE Matérn model with barriers. soap = Soap film smoother. te = Tensor product smoother. s = Bivariate and isotropic thin plate regression spline.

| Year | Model | $\hat{N}$ in 2017 Strata | | $\hat{N}$ in 2022 Strata | |
| | | Uncorrected | Corrected | Uncorrected | Corrected |
| --- | --- | --- | --- | --- | --- |
| 2017 | SPDE | 10140 | 11242 (0.111) | | |
| 2017 | soap | 10445 | 11665 (0.114) | | |
| 2017 | te | 10586 | 11963 (0.112) | | |
| 2017 | s | 10313 | 11747 (0.112) | | |
| 2017 | Ensemble | | 11654 (0.115) | | |
| 2017 | Conventional | 12269 (0.118) | | | |
| 2022 | SPDE | 7871 | 9446 | 10250 | 12023 (0.172) |
| 2022 | SPDE with barriers | 7936 | 10282 | 9598 | 12325 (0.198) |
| 2022 | soap | 9311 | 11980 | 12521 | 15593 (0.174) |
| 2022 | Ensemble | | | | 13313 (0.216) |
| 2022 | Conventional | 11891 (0.318) | | 19811 (0.343) | |

**Table 2  Number of random effects (#RE) used in the density surface models fitted to 2017 and 2022 Eastern Bering Sea beluga aerial line-transect survey data and included in the ensemble model for each year.** For a given model type, the number of random effects differs between years due to differences in the sample sizes available for fitting the models and in the study area extent. The effective degrees of freedom (EDF) were estimated using the mgcv package and are shown for all models except the SPDE Matérn with barriers, because the computation was not yet available for this model type. The percent deviance explained (Pct. Dev. Expl.) for each candidate model in the ensemble is also shown. Models showing "NA" in the table were not included in the ensemble model for that year.

| Model | 2017 | | | 2022 | | |
| | # RE | EDF | Pct. Dev. Expl. | # RE | EDF | Pct. Dev. Expl. |
| --- | --- | --- | --- | --- | --- | --- |
| SPDE Matérn | 199 | 39.2 | 53.6 | 308 | 50.1 | 83.4 |
| SPDE Matérn with barriers | NA | NA | NA | 316 | NA | 82.0 |
| Soap film smoother | 199 | 35.6 | 52.7 | 308 | 46.5 | 84.5 |
| Tensor product smoother | 195 | 40.2 | 56.3 | NA | NA | NA |
| Bivariate isotropic thin plate regression spline | 199 | 49.3 | 58 | NA | NA | NA |

Matérn with and without barriers, and soap film smoother (Fig. 3). The number of random effects used to fit each candidate model in the ensembles for 2017 and 2022 are shown in Table 2, along with the estimated effective degrees of freedom (computed using the mgcv package). The percent deviance explained is also shown in Table 2, and it was computed as $100*(1-\frac{R1}{R0})$, where $R1$ is the sum of squared deviance residuals for model $m$ and $R0$ is the sum of squared deviance residuals for the null (intercept-only) model. The candidate DSMs for 2017 explained between 52.7% (soap film smoother) and 58.0% (bivariate isotropic tprs) of the deviance. Among the 2022 candidate DSMs, the percent deviance explained ranged from 82.0% (SPDE Matérn with barriers) to 84.5% (soap film smoother).

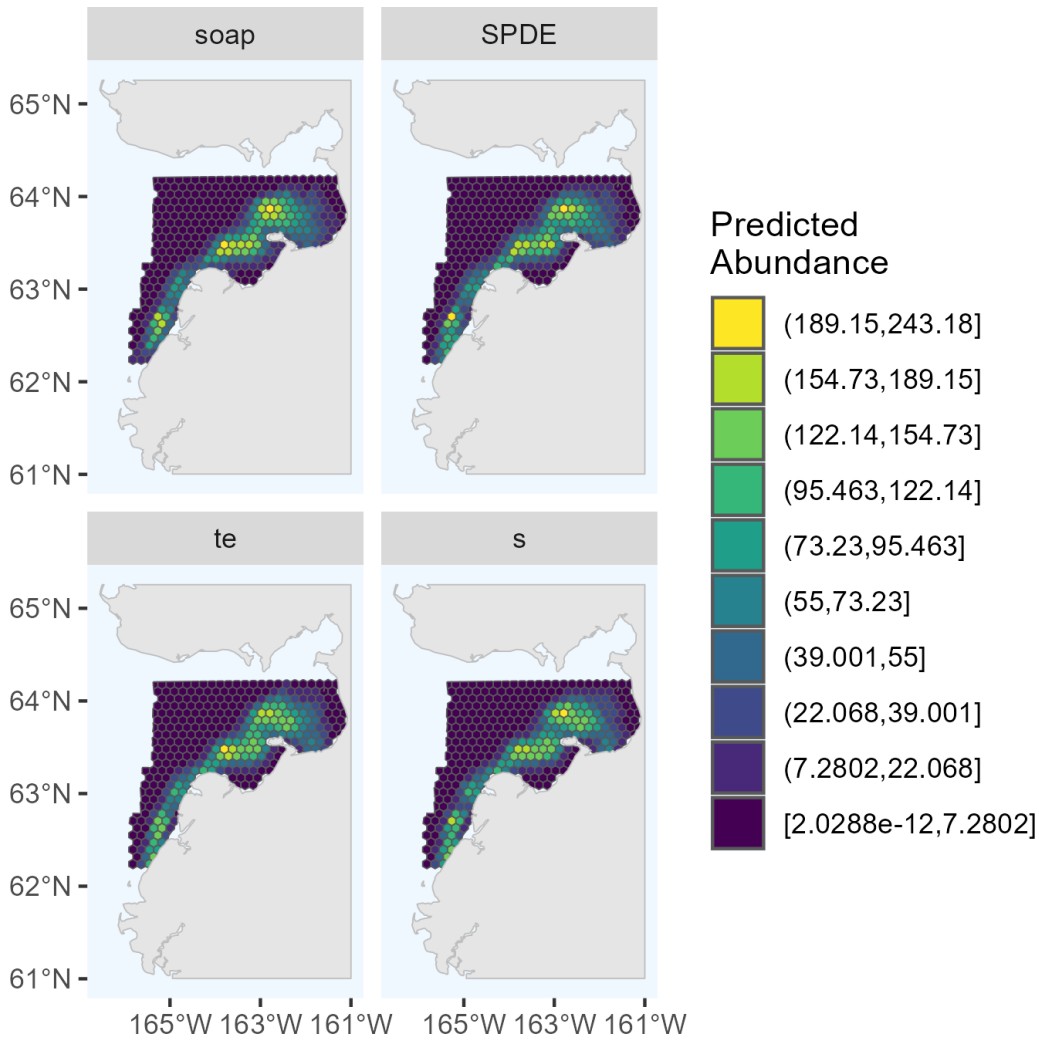

**Figure 2  Predicted abundance of Eastern Bering Sea belugas in 2017 based on the four candidate density surface models selected for the ensemble model.** soap, Soap film smoother; SPDE, SPDE Matérn model; te, Tensor product smoother; s, Bivariate and isotropic thin plate regression spline.

The maximum predicted abundance per cell was lower in 2017 than 2022. The year 2022 had less transect effort and fewer beluga sightings than 2017, and it also had three sightings with very large group sizes. Data with these characteristics can be challenging to fit with spatial models. To investigate differences among the 2022 DSMs, we conducted pairwise comparisons of predicted beluga abundance for all pairs of DSMs selected for the ensemble model. Specifically, for each cell $h$, we computed scaled differences in predicted abundance between models $m_1$ and $m_2$ as $(\hat{N}_{m_1,h} - \hat{N}_{m_1,h})/max(abs(\hat{N}_{m_1,h} - \hat{N}_{m_1,h}))$, where the denominator of this expression ensures that the resulting values are constrained to $[-1,1]$. Two points are worth highlighting from these comparisons (Fig. 4). First, predicted abundance was largely consistent between models throughout the overwhelming majority of the study area. The largest discrepancies in model predictions were in the area of high

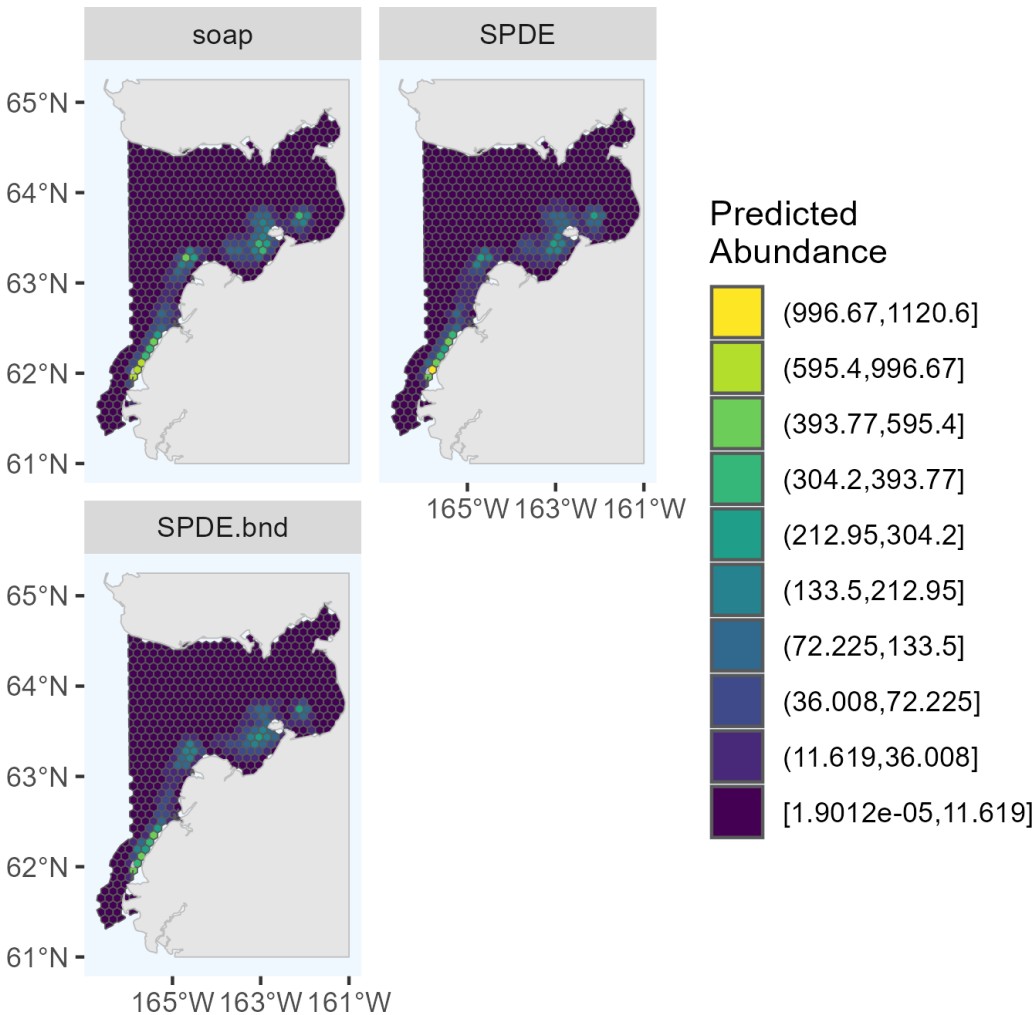

**Figure 3** **Predicted abundance of Eastern Bering Sea belugas in 2022 based on the three candidate density surface models selected for the ensemble model.** soap, Soap film smoother; SPDE, SPDE Matérn model; SPDE.bnd, SPDE Matérn model with barriers.

beluga sighting density near Scammon Bay, where some of the largest beluga groups were detected in 2022 (Fig. S2.2): the soap film smoother tended to estimate higher abundance than either SPDE Matérn-type model in a small cluster of cells near Scammon Bay; the SPDE Matérn model without barriers estimated higher abundance than the SPDE Matérn model with barriers in one cell.

The Tweedie distribution is underdispersed relative to the Poisson distribution when the mean is small; however, ecological data are often overdispersed. We expected that the EBS beluga data would be overdispersed. We examined whether the DSMs resulted in underdispersion by computing the ratio of the estimated variance to the estimated mean for each cell in the study area. The Tweedie parameter estimates for each of the cadidate DSMs included in the ensemble models are shown in Table 3. In 2017, the Tweedie parameter estimates were underdispersed only in cells in the northwestern corner of the study area,

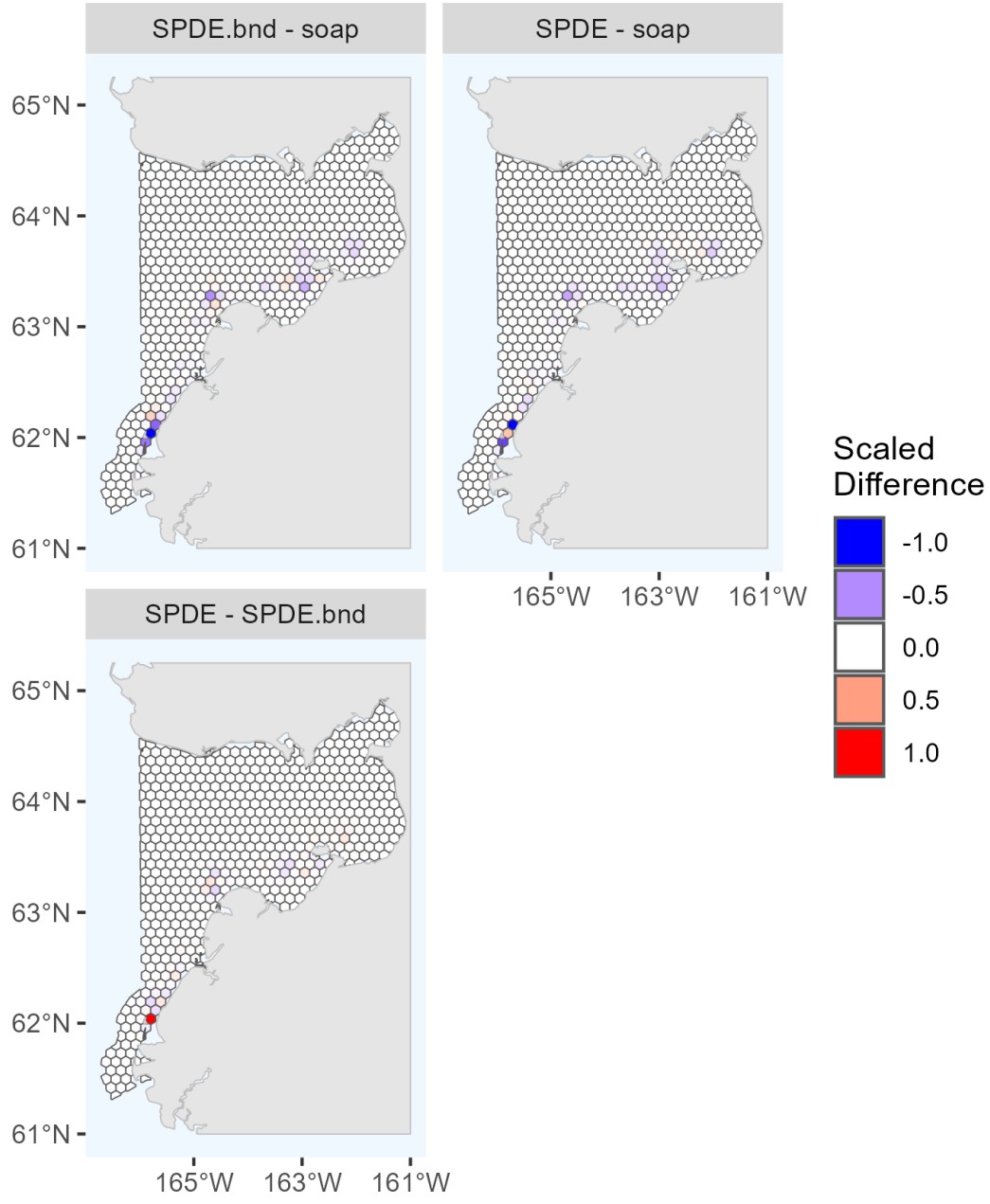

**Figure 4 Pairwise comparisons of density surface model (DSM) predictions for Eastern Bering Sea belugas in 2022.** Each map shows the scaled differences between two DSMs ($m_1$, $m_2$) in predicted beluga abundance by cell ($h$): $(\hat{N}_{m_1,h} - \hat{N}_{m_1,h})/max(abs(\hat{N}_{m_1,h} - \hat{N}_{m_1,h}))$. Comparisons between all pairs of DSMs selected for the ensemble model are shown. soap, Soap film smoother; SPDE, SPDE Matérn model; SPDE.bnd, SPDE Matérn model with barriers.

where no belugas were sighted and predicted densities were extremely low (S6). In 2022, all of the Tweedie parameter estimates were overdispersed.

The area-integrated estimates of EBS beluga abundance (with and without detransformation bias correction *via* the epsilon method) are shown in Table 1 and

**Table 3** Estimates of Tweedie dispersion ($\phi$) and power ($\rho$) parameters for density surface models fitted to Eastern Bering Sea beluga aerial line-transect survey data, where $Var(Y) = \phi\mu^\rho$.

| Year | Model | $\phi$ | $\rho$ |
|------|-------|--------|--------|
| 2017 | SPDE | 5.80 | 1.42 |
| 2017 | Soap film smoother | 5.85 | 1.42 |
| 2017 | Tensor product smoother | 5.58 | 1.42 |
| 2017 | Bivariate isotropic thin plate regression spline | 5.50 | 1.42 |
| 2022 | SPDE | 5.26 | 1.40 |
| 2022 | SPDE with barriers | 5.84 | 1.44 |
| 2022 | Soap film smoother | 5.08 | 1.41 |

Fig. 5 for the following: each of the candidate DSMs included in the 2017 and 2022 ensemble models; the ensemble models; and the conventional (post-stratified design-based) estimator. For the candidate DSMs in 2017, epsilon bias correction resulted in abundance estimates that were 11% to 14% larger than the plug-in estimators. For the candidate DSMs in 2022, the corresponding increase was higher, ranging between 17–28%. In 2017, the epsilon-corrected area-integrated abundance estimates ranged from 11,242 to 11,963 (CV = 0.111 to 0.114). In 2022, the range was from 12,023 to 15,593 (CV = 0.172 to 0.198). Nevertheless, within a survey year, the 95% lognormal confidence intervals for abundance overlapped across all candidate DSMs (Fig. 5). The ensemble spatial models estimated that there were 11,654 belugas in 2017 (CV = 0.115) and 13,313 belugas in 2022 (CV = 0.216). For comparison, the conventional, design-based models estimated that there were 12,269 belugas in 2017 (CV = 0.118; *Ferguson et al., 2023*) and 19,811 belugas in 2022 (CV = 0.343).

The abundance estimates for the full 2022 study area that were derived from two of the three candidate DSMs, the ensemble model, and the conventional estimator were all higher than the estimates for 2017. To investigate how much of this larger abundance in 2022 was due to the larger study area, we used each individual candidate DSM from 2022 to compute area-integrated abundances for the area corresponding to the geographic strata from the 2017 analysis. The results of this investigation suggest that approximately the same number of belugas were within the reduced study area during surveys in both years (Table 1).

## DISCUSSION

In this paper, we present detailed methods for constructing and evaluating hierarchical spatially explicit density models to estimate abundance from line-transect survey data using two leading model frameworks, SPDE approximations to geostatistical models and spline-based smoothers. Critical issues that we addressed include: (1) accounting for the precision and bias of all components of the hierarchical model in the final estimate of uncertainty in abundance *via* a parametric bootstrap; (2) applying the epsilon bias correction factor (*Thorson & Kristensen, 2016*) to account for detransformation bias in the DSM; (3) implementing a thorough model evaluation and selection process that incorporated examination of conditional PIT residuals, maps of model predictions, and extrapolation diagnostic metrics; and (4) using ensemble model averaging techniques to

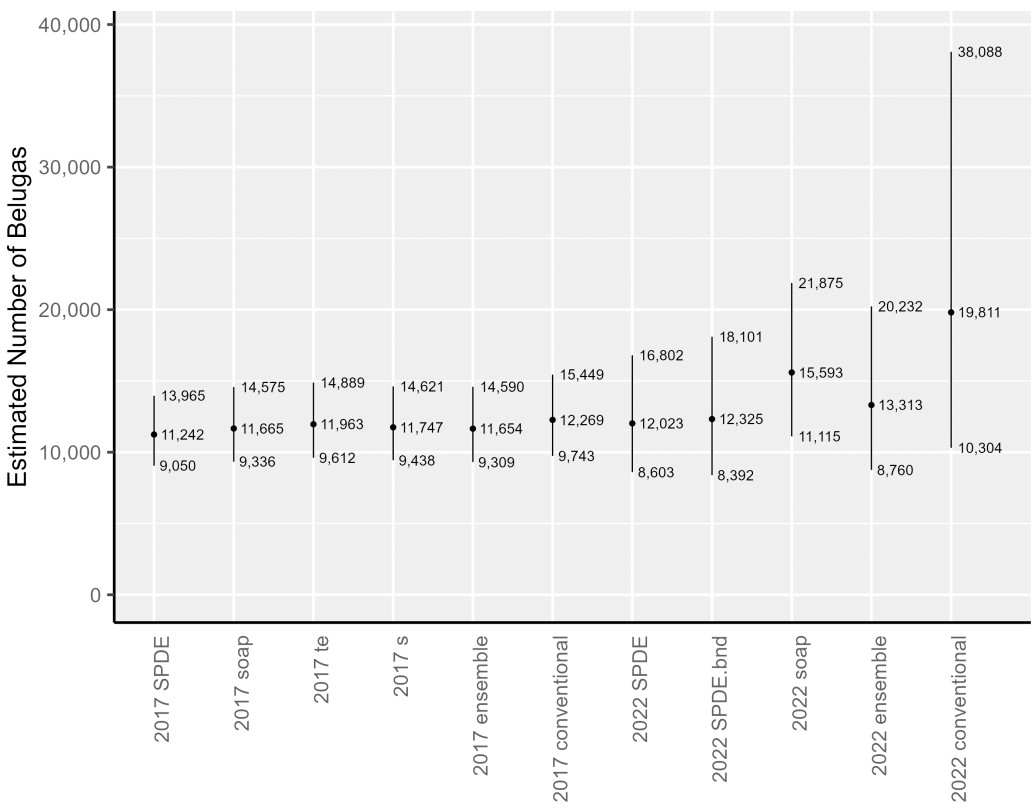

**Figure 5** **Estimates of abundance and lognormal 95% confidence intervals for Eastern Bering Sea belugas in 2017 and 2022.** SPDE, SPDE Matérn model; SPDE.bnd, SPDE Matérn model with barriers; soap, Soap film smoother; s, Bivariate and isotropic thin plate regression spline; te, Tensor product smoother.

derive the ultimate estimates of abundance and uncertainty to account for model selection uncertainty, which is especially important in situations for which different structurally sound models produce different results.

We demonstrated our methods using Eastern Bering Sea belugas as a case study. This was a particularly informative case study because aerial line-transect surveys were conducted using identical protocols during two years in which survey effort and beluga distribution differed dramatically, 2017 and 2022. The goals of the EBS beluga analysis were to (1) assess whether DSMs represent an improvement over conventional, post-stratified design-based estimators; (2) produce updated estimates of abundance for this stock; and (3) produce detailed maps of beluga density in the survey area during the 2017 and 2022 survey periods. For the case study, we constructed identical DSMs using two alternative methods for building basis functions representing spatially correlated variation in population density; that is, the SPDE approximation to the Matèrn covariance function and bivariate splines.

For EBS belugas, DSMs produced similar estimates to the conventional design-based estimator in 2017, and were considerably lower than the design-based estimator in 2022 (Table 1). The estimated precision in the abundance estimates from the individual DSMs and the ensemble average of DSMs was higher (*i.e.,* lower CVs) than for the design-based

estimator in both survey years (Table 1). The decreased precision in all of the abundance estimates for 2022 compared to 2017 is likely a function of both decreased survey effort and increased beluga clustering, particularly in the southern portion of the survey area that had not be covered by aerial line-transect surveys for belugas in the past. Increased survey effort, redistribution of survey effort, or some combination thereof will be needed in future surveys to reduce uncertainty in the abundance estimate.

The abundance estimates for the 2017 study area are similar across estimators (Fig. 5) and between survey years (Table 1, Fig. 5). The point estimates of abundance in the full 2022 study area for two of the three candidate DSMs included in the 2022 ensemble model average (*i.e.,* the SPDE Matérn and SPDE Matérn with barriers) are within the 95% confidence intervals for all of the 2017 abundance estimates (Fig. 5), which correspond to the smaller study area. The point estimate for the 2022 soap film smoother is only slightly larger than the upper 95% confidence interval for the 2017 models. The ensemble model average abundance estimate for the smaller study area in 2017 (11,654 belugas, CV = 0.115) is only 1,659 belugas lower than the ensemble abundance estimate for the larger study area in 2022 (13,313 belugas, CV = 0.216). This difference is approximately 12.5% of the ensemble abundance estimate for 2022. The slightly larger abundance estimates in 2022 could be due to a number of factors, including random error, actual population decrease or increase between years, emigration or immigration from the system, and increased survey area between 2017 and 2022.

For 2022, the conventional abundance estimate (19,811 belugas, CV = 0.343) is approximately 49% larger than the corresponding ensemble abundance estimate (13,313 belugas, CV = 0.216). We cannot know for certain whether the spatially explicit estimators or the design-based estimator provides the least biased estimate of abundance for the area surveyed. However, based on the following information, we believe that the spatially explicit models are the least biased for this case study. The three extremely large groups of belugas (67, 87, and 120 belugas) detected in 2022 were located nearshore, north of Scammon Bay, in the new southern portion of the study area that was never surveyed prior to 2022 (Fig. S2.2). The 2022 design-based estimator appears to be spreading the influence of the three large groups over a broad area, resulting in a high estimate of $\hat{N}$ for the stratum. It is unknown whether there were many undetected large groups in unsurveyed locations (*i.e.,* between transects within the existing survey area boundaries); however, we believe that it was unlikely that many large groups were missed. The southern extension of the study area was surveyed completely in 2022, according to the survey design, and the transects were surveyed in Beaufort Sea State 2 (Fig. S2.3). A preliminary soap film smoother model for 2022 that was constructed using fewer knots and only 146 random effects (compared to 308 random effects in the present analysis; Table 2) resulted in a considerably larger bias-corrected abundance estimate (20,162 belugas, CV = 0.26; compared to 15,593 belugas, CV = 0.174, in the present analysis; Table 1) and larger estimates of the Tweedie dispersion ($\phi$, 7.42 *vs.* 5.08; Table 3) and power ($\rho$, 1.49 *vs.* 1.41; Table 3) parameter estimates. This suggests that the DSMs with the larger number of spatial random effects that were used in the present analysis were better able to represent the patchy beluga sightings in the data.

Based on the information presented above, we believe that the bias-corrected ensemble model average abundance estimate derived from the spatially explicit models is less biased than the conventional estimate for 2022. Furthermore, because the density surface modeling paradigm also enables estimation of higher resolution maps of species density (Figs. 2 and 3), we view DSMs as an improvement in statistical methodology for analyzing EBS beluga data to maximize utility in management and conservation decisions. We recommend the 2022 ensemble abundance estimate as the most pertinent for management at present, recognizing that there are still some unaddressed issues (*e.g.*, no sampling of belugas in rivers; M Ferguson with M Castellote, pers. comm., 2023) that likely make it an underestimate.

Our study offers a number of lessons for researchers seeking to implement DSMs, whether with beluga or other species. First, it was apparent from our analysis of the 2022 data that models with different spatial basis function formulations have the potential to produce quite different abundance estimates (*Ferguson, Conn & Thorson, 2024*). This is likely due to the way in which estimated abundance is interpolated (and extrapolated) into unsampled areas. The tendency for this to occur may be affected by sampling intensity (lower in 2022) and the level of overdispersion (several large group sizes in 2022). Therefore, we strongly caution against employing just one form of spatial model. Instead, we recommend that investigators routinely fit models with different spatial basis functions, implement thorough and repeatable model evaluation and selection process, and consider ensemble modeling (*Araújo & New, 2007*) if models produce substantially different predictions.

Our approach in this paper was to employ ensemble models with equal weighting. Alternatives, such as using an information criterion (*Burnham & Anderson, 2002*) to weight models, are certainly possible. To our knowledge, the performance of marginal AIC (*i.e.,* ignoring spatial random effects when counting parameters) for model weighting has not been rigorously evaluated. In our case, using marginal AIC to weight models would have placed virtually all model weight on the soap film smoother model for 2022. To be conservative, we thus adopted an equal weighting strategy, which has been shown to be reasonable in practice (*Dormann et al., 2018*). An equal weighting strategy does require some forethought in developing a "balanced" model set, so that a particular class of models (*e.g.*, with minor permutations in structure) does not dominate inference (*Dormann et al., 2018*). In the EBS beluga case study, the full suite of candidate models we evaluated included different boundary conditions and spatial autocorrelation characteristics (*i.e.,* with and without barriers; isotropic and anisotropic), and covered a range of model complexity, measured by the number of random effects and effective degrees of freedom. Alternative strategies for DSM ensemble weighting would make for useful future research, potentially using recent advances in computing cAIC for the spatial models used here (*Zheng, Cadigan & Thorson, 2024*).

Although we recommend spatial DSMs for estimating abundance of EBS belugas, this is not a disavowal of general principles of survey design. Such principles (*e.g.*, randomization and replication; *Buckland et al., 2001*) help to ensure that model-based estimators will be unbiased and should be regarded as good practice in transect surveys, no matter the

method used to analyze the data (*Hedley & Bravington, 2014*). Design-based concepts in survey design (*e.g.*, systematic random samples) are still important for the quality of inference in DSMs.

## CONCLUSIONS

Density surface models (DSMs) are commonly fitted to counts obtained during line-transect surveys of marine mammal populations as an alternative to conventional design-based estimators. For EBS belugas, we found DSMs to be preferable, given the extra information one gains through maps of spatial distributions. However, when fitting DSMs, researchers need to be cognizant that different spatial basis functions can result in different estimates, particularly when animals are patchily distributed. In such cases, use of ensemble predictions are likely warranted. Further, investigators should take care to properly account for uncertainty by propagating uncertainty in detection probability into resultant estimates, and to account for possible detransformation bias.

## ACKNOWLEDGEMENTS

We thank the hunters and scientists who provided invaluable knowledge and support for the Eastern Bering Sea beluga aerial line-transect surveys in 2017 and 2022. The Indigenous knowledge that the ABWC provided was invaluable to identifying the appropriate study area boundaries. The aerial survey observers included A. Brower (2017 and 2022), K. Shelden (2017 and 2022), C.Sims (2017), and M. Ferguson (2022). Aircraft support in both years was provided by Clearwater Air, Inc.; we appreciate the dedication and skills of the Clearwater pilots A. Harcombe, J. Turner, C. Wilson, S. Corbin, K. Vacendak, and D. McDonald. H. Skaug and D. Miller provided detailed, functional, and educational examples of DSMs in TMB and R that formed the basis for all the models evaluated in this paper. We thank A. Warlick, J. Ver Hoef, J. Lee, H. Skaug, D. Miller, T. Doniol-Valcroze, and an anonymous reviewer for comments on previous versions of this manuscript that enhanced its quality.

### Funding

Funding for this work was provided by the National Oceanic and Atmospheric Administration and the Alaska Beluga Whale Committee. The Alaska Beluga Whale Committee assisted with designing the aerial line transect surveys conducted in 2017 and 2022. Two authors are currently employed by NOAA. The lead author was employed by NOAA when the field work was conducted and for a portion of the data analysis period. The scientific results and conclusions, as well as any views or opinions expressed herein, are those of the authors and do not necessarily reflect the views of NOAA or the Department of Commerce.

## Grant Disclosures

The following grant information was disclosed by the authors:
National Oceanic and Atmospheric Administration and the Alaska Beluga Whale Committee.

## Competing Interests

Paul B. Conn and James T. Thorson are employed by National Oceanic and Atmospheric Administration. Megan C. Ferguson was employed by the National Ocean and Atmospheric Administration from 2008 to September 2023, and is now employed by the Biodiversity Research Institute.

## Author Contributions

- Megan C. Ferguson conceived and designed the experiments, performed the experiments, analyzed the data, prepared figures and/or tables, authored or reviewed drafts of the article, and approved the final draft.
- Paul B. Conn analyzed the data, authored or reviewed drafts of the article, and approved the final draft.
- James T. Thorson analyzed the data, authored or reviewed drafts of the article, and approved the final draft.

## Animal Ethics

The following information was supplied relating to ethical approvals (i.e., approving body and any reference numbers):

The aerial survey research was approved by the Alaska Fisheries Science Center/Northwest Fisheries Science Center Institutional Animal Care and Use Committee.

## Data Availability

Data and code are available on GitHub and Zenodo:

- Available at https://github.com/meganferg/FergusonEtal_20250125_EBS_Beluga_DSM.

- Ferguson, M. (2025). FergusonEtal_20250125_EBS_Beluga_DSM. Zenodo. Available at https://doi.org/10.5281/zenodo.16269136.

## Supplemental Information

Supplemental information for this article can be found online at http://dx.doi.org/10.7717/peerj.20077#supplemental-information.

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
