# Peer review of "Spatially explicit models of density improve estimates of Eastern Bering Sea beluga (*Delphinapterus leucas*) abundance and distribution from line-transect surveys"

_PeerJ, doi:10.7717/peerj.20077_

## Round 0.1 · original submission · Minor Revisions

Both reviewers commend the manuscript's clear writing, comprehensive nature, and timely contribution to the field of abundance estimation using spatially-explicit methods. They highlight the importance of your work for both ecological research and applied management, particularly your rigorous comparison of design-based and model-based estimators and your exploration of model ensemble approaches. The availability of your data and code is also greatly appreciated, enabling full replication.

While both reviewers recommend publication, they have also provided valuable feedback and suggestions for minor revisions that will further strengthen the manuscript.

Please address the following points in your revised submission:
1. Reference to Non-Peer-Reviewed Material and missing references
2. Model Averaging and Equal Weighting
3. Inclusion of Transect Detection Probability and Variance
4. Penalization for Matérn Models

Reviewer 1 ·

Basic reporting

The manuscript is very well written and requires only minor revisions prior to publication. The background material is both important and well characterized. Regarding the references, a few of the papers cited are not generally considered peer reviewed (e.g., papers submitted as IWC SC documents). Perhaps they should be referenced as footnotes, depending on journal policy. Also, the reference list was missing a reference to Dorfman 1938. The tables and figures are excellent and all should be included in the publication. Finally, it is my understanding from the text that the raw data are available upon request.

Experimental design

This is a very technical manuscript, which includes considerable detail on protocols for estimating abundance with design based and model based estimators. I believe sufficient details were provided relative to the methods used, although this is not my area of expertise. The fundamental research question posed in this manuscript is an important issue for both ecologists and the applied research community. Setting “take” or “harvest” limits proportional to estimates of abundance is considered a reasonable standard in management. Given the well documented approaches to incorporating uncertainty in the abundance estimates, appropriately precautionary management strategies can be based on the information provided herein.

Validity of the findings

In my opinion the approach is robust and statistically sound. While highly technical, the manuscript provides sufficient detail to duplicate the analysis, as well as a link to acquire the data used by the authors. The authors provide a compelling rationale for scientists to use model based estimators of abundance, and where appropriate, ensemble estimates based on a series of model based estimators. In addition, the authors provided an excellent analysis comparing traditional line transect estimates of abundance (design based) with the spatially integrated (model based) estimates of abundance. This is an important paper, and in my opinion, should be considered a significant contribution to the published literature on this subject.

Additional comments

Specific comments:
Line 69 – change “stock of belugas” to “stock of beluga” or “stock of beluga whale”.
Line 74 – change “NOAA” to “NMFS” (or perhaps “NOAA/NMFS”, as the Adams et al. reference refers to NMFS, not NOAA, and the ABWC Management Plan refers to “NOAA/NMFS” and “NMFS Enforcement”.
Line 102 – change “NOAA” to “NMFS”, as the collaboration was with NMFS staff and the ABWC.
Line 633- change “149% larger” to “49% larger”.
Line 659 – delete “slight”.
Line 661 – change “belugas” to “beluga” or “beluga whale”.
Line 678 – insert “for estimating abundance of” before “EBS belugas”.
Line 682 – insert “,” after “e.g.”.
Lines 702/3 – perhaps delete the last sentence of the Acknowledgements.
Line 733 – there was a reference to Dorfman (1938) that should be included in the citation (unless it was in error).
Lines 744/5/6/ - perhaps this reference should be added as a footnote. It is not a peer-reviewed paper, by most definitions of “peer review”.
Lines 747/8/9 – same concern. Should only peer reviewed articles be included in this section? Same comment for other papers submitted to the IWC Scientific Committee.

·

Basic reporting

This study provides a timely and relevant description of spatially-explicit methods for estimating population abundance from line-transect surveys. It builds on previous publications about estimating and propagating uncertainty in density surface models (e.g., Miller et al. 2022) and on the differences and similarities between SPDE approaches and spline smoothers (Miller et al. 2020), with a focus on concrete applications and comparisons using realistic situations and datasets. The manuscript is written in a clear, precise and concise language, and demonstrates a thorough grasp of the relevant literature as well as the management background of its applied example. Its structure is well laid-out from the start and very clear in distinguishing and explaining the different parts of the complex analysis workflow. The inclusion of a case-study that illustrates the methods and main points of discussion while also providing important real-life information for management purposes makes it a particularly useful and self-contained paper.

Experimental design

The study clearly defines its main research questions and tackles them with a rigorous review and application of state-of-the art density estimation techniques, including new methodological contributions. The supplemental material and the availability of the data and code in a public GitHub repository allows for full replication of the methods. The design and methods of the aerial surveys themselves are standard and well-proven.

Validity of the findings

A. Model averaging and equal weighting
The authors do the field a service by acknowledging that “There are a considerable number of ways to formulate DSMs” (line 110) and stating that they are “particularly interested in the sensitivity of abundance estimates to DSM model structure” (lines 65-66) in case “different models produce markedly different estimates” (line 420). They propose tackling this issue by using multiple rigorous criteria to evaluate and select their candidate model set (lines 354-358) and using a model ensemble approach to produce the final estimates. Their case-study gives the reader a sense of the impact that the choice of a particular model or modelling approach (e.g., SPDE vs splines smoothers) can have on the abundance estimate: in this case, Table 1 shows that the 2017 estimates yielded by 1 SPDE model and 3 spline models (1 soap, 1 te, 1 s) are all within a few percentages of each other (11,242 – 11,963). In 2022, however, the 2 SPDE estimates are very close (12,023, 12,325) but the soap smoother yields an abundance estimate that is 25% larger (15,593).

The authors state that “The advantage of averaging models is that there is often a reduction in prediction error” (line 424) but I feel an even more important message here is that the ensemble approach will account appropriately for the additional uncertainty generated by different models producing different estimates. One tricky question, however, is choosing how models should be weighted. The authors cite Dormann et al. (2018) to justify the use of equal weights, arguing that relying on calculated weights (e.g. using AIC) can be computationally complex for hierarchical models and often result in a single model dominating the inference. This is a defensible position, but I am concerned about how to ensure that the final model set is well balanced and how to avoid the mean estimate being influenced by multiple similar models. This issue is recognized in Dormann et al. (2018): “Statistical models, which aim to describe the data to which they are fitted, will often have correlated parameters and fits (…) Having highly similar models in the model set will inflate the cumulative weight given to them. (…) Another approach would be to pre-select models of different types.”

In the example of the 2022 estimates, one could argue that the inclusion of 2 SPDE models with almost identical estimates and only 1 spline model effectively pulls the weighted average towards the lower estimates. All of those models are valid, but presumably one could find additional models that would also be valid and would differ in other aspects of the parametrization (this would be especially true with models using multiple combinations of environmental covariates, as is common in this field). How then can one avoid creating (not necessarily on purpose) an unbalanced set of models where multiple similar models have a disproportionate influence (what Dormann et al. call “reinforce the trend of emphasizing the model type most represented in the set”)?

I do not necessarily have concerns about the specific results for the beluga surveys in this manuscript. However, since the authors have chosen to present their work (very usefully) as a review and comparison of different analytical methods, the paper can and will be used as a ‘recipe’ for this type of analysis, and therefore extra caution should be taken to avoid prescribing a single approach (equal weights) without discussing more fully its risks and benefits. To this end, I would recommend expanding the relevant parts of the methods (426-434), results, and discission (669-677) to better acknowledge the issue and how it could affect the results of this specific study (note for instance that line 673 states that “using marginal AIC to weight models would have placed virtually all model weight on a single model for 2022” but does not tell us which one or discuss the impact), and perhaps provide recommendations on how to achieve ‘balanced’ sets of candidate models.

Additional comments

B. Inclusion of transect detection probability and variance: S3 summarizes how observer and imagery data from previous surveys were used to derive a detection probability at the transect line, with the details available in previous publications. The use of those data is well justified and reasonable, but the actual pMR (0,zj;θ MR) is not given in S3 nor in the body of the manuscript in either the methods or the results. It would be good to provide the value of both the perception and the availability corrections in the results to help the readers assess the influence of those corrections on the resulting abundance estimates.

Also, lines 436-438 only mention the variance of the perception correction factor but not that of the availability bias because, in this particular case, there is no estimate of uncertainty for availability probability (line 445). This makes the lines 436-438 specific to this study even though they are placed in the ‘general’ description of the methods and workflow. I recommend making lines 436-438 more general (i.e., mention that the delta-method can be used to add the uncertainty components for both perception and availability biases) and then use the following section to explain that the case-study only includes one of those components.

C. Some of the deviance explained for 2022 Matérn models is quite high (>80%, Table 2), which suggests good fit but could also be a sign of overfitting to the data. It is clear from the methods how the spline-based smoothers are penalized to avoid overfitting (e.g., lines 308-313) but in the spirit of the state objective to “identify similarities and differences among different analytical approaches”, can you comment briefly on how equivalent penalization is achieved with the Matérn models? (Recognizing that since the deviance explained for the 2017 results is similar between the Matérn and the spline models, the two methods seem to yield a similar degree of smoothing.)

---

## Round 0.2 · accepted · Accept

Congratulations on the great work!